# Water-assisted oxidative redispersion of Cu particles through formation of Cu hydroxide at room temperature

Yamei Fan[1,2,5], Rongtan Li[2,5], Beibei Wang[3], Xiaohui Feng[1,2], Xiangze Du[2], Chengxiang Liu[2], Fei Wang[4], Conghui Liu[2], Cui Dong[2], Yanxiao Ning[2], Rentao Mu[2] & Qiang Fu[2] ✉

Sintering of active metal species often happens during catalytic reactions, which requires redispersion in a reactive atmosphere at elevated temperatures to recover the activity. Herein, we report a simple method to redisperse sintered Cu catalysts via $O_2$-$H_2O$ treatment at room temperature. In-situ spectroscopic characterizations reveal that $H_2O$ induces the formation of hydroxylated Cu species in humid $O_2$, pushing surface diffusion of Cu atoms at room temperature. Further, surface OH groups formed on most hydroxylable support surfaces such as γ-$Al_2O_3$, $SiO_2$, and $CeO_2$ in the humid atmosphere help to pull the mobile Cu species and enhance Cu redispersion. Both pushing and pulling effects of gaseous $H_2O$ promote the structural transformation of Cu aggregates into highly dispersed Cu species at room temperature, which exhibit enhanced activity in reverse water gas shift and preferential oxidation of carbon monoxide reactions. These findings highlight the important role of $H_2O$ in the dynamic structure evolution of supported metal nanocatalysts and lay the foundation for the regeneration of sintered catalysts under mild conditions.

Supported metal nanocatalysts are commonly employed in heterogeneous catalysis, while sintering of supported metal species often happens during high-temperature reactions leading to catalyst deactivation[1–5]. To address this issue, various redispersion strategies have been developed with the aim of reversing the sintering and revitalizing the active metal species[6–11]. The redispersion of metal species typically involves detachment of metal atoms from larger particles, followed by surface or vapor-phase migration and final capture by surface anchoring sites[10,12]. The success of redispersion largely depends on metal-support interaction, determining thermodynamic favorability of the process[8–10,13–15]. Therefore, supports with various surface defects or surface functional groups (oxygen vacancy, OH, heteroatom, etc.) are essential for anchoring mobile metal species

through strong interaction between metal and support[8,9,13,14]. Furthermore, kinetic factors also play a pivotal role which cannot be underestimated. Thus, high-temperature treatments in specific gaseous environments ($CH_3I$, $O_2$, $NH_3$, etc.) are often employed to enhance the mobility of metallic species across the support surface or through gas-phase migration[7,16–18]. Apparently, these redispersion processes require a considerable energy input. The quest for eco-friendly and energy-saving redispersion strategies remains an urgent priority[10].

Owing to the low melting temperature of copper metal (1083 °C) and its low Hüttig and Tammann temperatures (174 and 405 °C, respectively), Cu-based catalysts are susceptible to sintering, thereby limiting their industrial applications[19,20]. However, the high mobility of Cu atoms also allows the facile redispersion of Cu particles under

[1]Department of Chemical Physics, University of Science and Technology of China, Hefei, China. [2]State Key Laboratory of Catalysis, Chinese Academy of Sciences, Dalian Institute of Physics, Dalian, China. [3]Center for Transformative Science, ShanghaiTech University, Shanghai, China. [4]Faculty of Environmental Science and Engineering, Kunming University of Science and Technology, Kunming, China. [5]These authors contributed equally: Yamei Fan, Rongtan Li. ✉ e-mail: qfu@dicp.ac.cn

relatively mild conditions[21,22]. In recent years, the structural change of Cu nanocatalysts at room temperature (RT) has been occasionally reported[23-28]. The incorporation of Cu atoms from Cu nanoparticles (NPs) into silica matrix[24], intercalation of Cu atoms from bulk Cu into layered transition metal dichalcogenides[25], and disintegration of Cu nanoparticles into single atoms on N-doped carbon supports[23,27] have been observed under ambient conditions, which are driven by coordination of Cu atoms with surface atoms of the support e.g., O, Si, S, and N. Sun et al. find that pre-adsorbed $H_2O$ on ZnO surface can also promote redispersion of Cu particle into single atoms and few-atom Cu clusters at RT. Despite these interesting findings the mechanism underlying the redispersion process and the dynamic interaction between gaseous atmosphere and metal atoms/support during the processes remain unclear, thus necessitating in-depth studies.

In this work, the dynamic behavior of supported Cu NPs in various atmospheres at RT has been investigated by a variety of characterization techniques including high-angle annular dark-field scanning transmission electron microscopy (HAADF-STEM), X-ray photoelectron spectroscopy (XPS), X-ray absorption spectroscopy (XAS), and ultraviolet-visible (UV-Vis) diffuse reflectance spectroscopy. The results reveal the spontaneous redispersion of aggregated Cu particles into Cu single atoms and ultrasmall clusters on $\gamma$-$Al_2O_3$ in humid air and at RT. $H_2O$ is found to promote the formation of mobile hydroxylated Cu species in $O_2$, pushing the diffusion of Cu atoms. Meanwhile, the enriched surface OH groups of the support in a humid atmosphere provide anchoring sites to pull the diffusing hydroxylated Cu atoms. Redispersion of Cu particles into Cu single atoms and clusters under mild ambient conditions are achieved by synergizing the thermodynamic and kinetic effects of $H_2O$. The generated Cu single atoms and clusters on $\gamma$-$Al_2O_3$ and $CeO_2$ supports exhibit high activity in reverse water gas shift (RWGS) and preferential oxidation of carbon monoxide (CO-PROX) reactions, respectively. Moreover, the deactivated Cu catalysts after the reactions can be facilely reactivated by exposing to $O_2$-$H_2O$ at RT.

## Results

### Spontaneous redispersion of Cu NPs in air at RT

$\gamma$-$Al_2O_3$ supports were prepared by calcination of pseudo-boehmite at different temperatures as confirmed by X-ray diffraction (XRD) in Fig. S1, which are denoted as AlOOH-$T$ ($T$ represents calcination temperature, $T$ = 500 and 900 °C)[14]. Cu NPs supported on $\gamma$-$Al_2O_3$ (2 wt.%) were prepared by wet impregnation method and denoted as 2Cu/AlOOH-$T$ (details seen in Methods). The absorption band around 570 nm in UV-Vis spectra (Fig. 1a) and diffraction peaks at 43.3 and 50.4° in XRD patterns (Fig. S2) are characteristic for metallic Cu NPs[29-31]. Interestingly, these signals for Cu NPs disappear after exposure to air for one week (Fig. 1a and Fig. S2). A new broad absorption peak around 600−800 nm assigned to $Cu^{2+}$ species[32] appears and no diffraction peaks of Cu species are present, indicating that Cu NPs may transform into smaller Cu species in air.

In-situ UV-Vis experiments were conducted to monitor the structural evolution of Cu species in 2Cu/AlOOH-900 when exposed to each component of air at RT including $N_2$, Ar, $CO_2$, $O_2$, Ar-$H_2O$ and $O_2$-$H_2O$. The fresh sample was used for each in-situ characterization. Figure 1a shows that the characteristic adsorption peak of metallic Cu NPs remains unchanged in $N_2$, Ar, and $CO_2$, yet is markedly diminished in $O_2$, Ar-$H_2O$ and $O_2$-$H_2O$ gases. The observation suggests that the redispersion of Cu NPs in air may be caused by the oxidizing gas components including $O_2$, $H_2O$, and both together.

It is known that atomically dispersed $Cu^{2+}$ species with an unpaired electron in the $d_{x^2-y^2}$ orbital are active for electron paramagnetic resonance (EPR) while $Cu^{2+}$ ions in crystalline CuO are EPR-inactive due to the strong antiferromagnetic coupling[33-36]. No obvious EPR signal is observed for 2Cu/AlOOH-900 (Fig. 1b), indicating the absence of highly dispersed $Cu^{2+}$ species in the fresh sample. In contrast, much strong EPR signals around 3300 G are detected in 2Cu/AlOOH-900 after treatment in $O_2$-$H_2O$ for 24 h at RT as well as in Ar-$H_2O$ and $O_2$ (Fig. 1b), implying that $O_2$-$H_2O$ treatment generates highly dispersed $Cu^{2+}$ species[35].

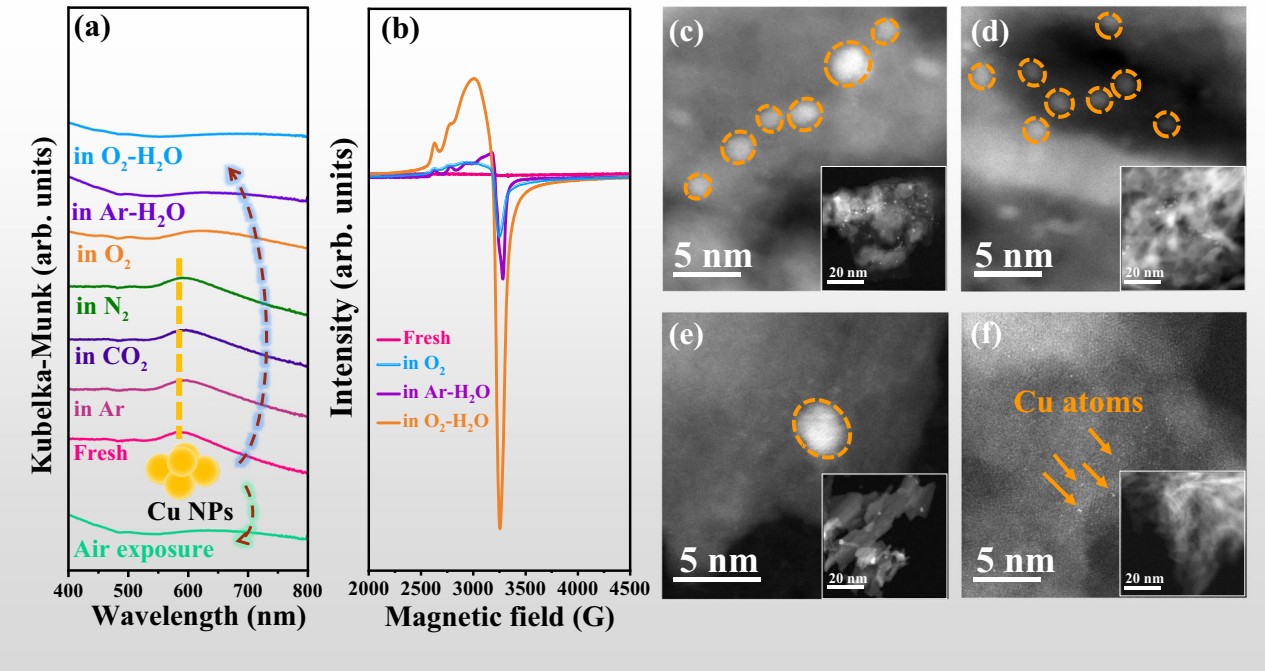

**Fig. 1 | Spontaneous redispersion of Cu NPs at RT. a** In-situ UV-Vis spectra of 2Cu/AlOOH-900 treated in different atmospheres (Air, Ar, $CO_2$, $N_2$, Ar-$H_2O$, $O_2$, and $O_2$-$H_2O$) for 30 min. **b** EPR spectra of 2Cu/AlOOH-900 before and after treatment in $O_2$, Ar-$H_2O$ and $O_2$-$H_2O$ atmospheres for 24 h. HAADF-STEM images with high-magnification and low-magnification (insets) of (**c**) 2Cu/AlOOH-900, 2Cu/AlOOH-900 treated in (**d**) $O_2$, (**e**) Ar-$H_2O$ and (**f**) $O_2$-$H_2O$ for 24 h.

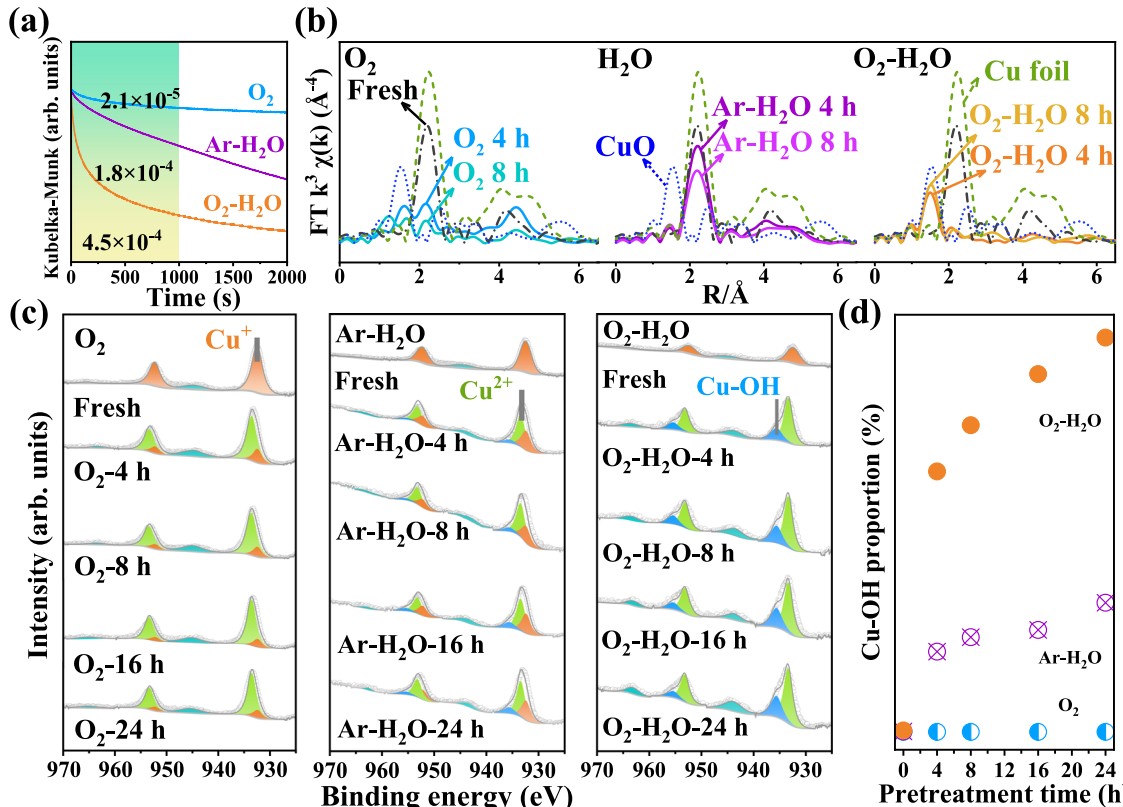

**Fig. 2 | $H_2O$ effect on redispersion of Cu NPs at RT. a** In situ UV-Vis spectra of 2Cu/AlOOH-900 treated in different atmospheres. **b** Quasi in-situ Fourier-transforms of $k^3$-weighted Cu K-edge EXAFS spectra of 2Cu/AlOOH-900 treated in $O_2$, Ar-$H_2O$ and $O_2$-$H_2O$ for 4 h and 8 h, as well as standard samples of Cu foil and CuO. **c** Quasi in-situ Cu 2p XPS spectra of 2Cu/AlOOH-900 treated in $O_2$, Ar-$H_2O$ and $O_2$-$H_2O$ for 4, 8, 16 and 24 h. **d** Changes in the proportion of Cu-OH species over time in different atmospheres calculated based on Cu 2p XPS spectra.

---

HAADF-STEM was then used to determine the size of Cu species treated in different atmospheres. Cu NPs with an average diameter of about 3 nm are observed in the fresh 2Cu/AlOOH-900 sample (Fig. 1c and corresponding inset). After treatment in $O_2$ and Ar-$H_2O$ at RT for 24 h, Cu clusters with size of 2 nm are still observed (Fig. 1d, e and corresponding insets). In contrast, only Cu single atoms are imaged in the $O_2$-$H_2O$ treated sample (Fig. 1f and corresponding inset). These results illustrate that Cu NPs are completely redispersed into Cu single atoms in an $O_2$-$H_2O$ atmosphere but only partly redispersed in $O_2$ or Ar-$H_2O$ at RT for 24 h.

### Effect of $H_2O$ on Cu redispersion process

Time-dependent in-situ UV-Vis spectroscopy was employed to monitor the evolution of Cu species in $O_2$, Ar-$H_2O$ and $O_2$-$H_2O$ atmospheres. The redispersion rate of metal NPs can be represented by the slope of the kinetic curve (-$\Delta$KM/$\Delta t$)[14,37]. As shown in Fig. 2a, the redispersion rate at early stage (1000 s) in various atmospheres follows the sequence of $O_2$-$H_2O$ ($4.5 \times 10^{-4}$) > Ar-$H_2O$ ($1.8 \times 10^{-4}$) > $O_2$ ($2.1 \times 10^{-5}$), indicating that $O_2$-$H_2O$ atmosphere accelerates the redispersion process. Subsequently, quasi in-situ XAS experiments were conducted to identify the chemical state of Cu in $O_2$, Ar-$H_2O$ and $O_2$-$H_2O$ atmospheres. As shown in Fig. 2b and Fig. S3, the main peak centered at 2.2 Å in extended X-ray absorption fine structure (EXAFS) spectra of 2Cu/AlOOH-900 is assigned to Cu-Cu bond similar to that in Cu foil, implying that metallic Cu dominates in the fresh sample, which agrees with the Cu K-edge X-ray absorption near-edge structure (XANES) results (Fig. 2b, Figs. S3 and S4)[38,39]. The peak of Cu-Cu bond disappears in $O_2$-$H_2O$ for 4 h while only gets weaker in $O_2$ and Ar-$H_2O$ even if the treatment time is extended to 8 h (best-fit parameters summarized in Table S1). Meanwhile, a peak around 1.5 Å appears (Fig. 2b and Fig. S3)

which is the typical scattering feature of Cu-O coordination, accompanied by the obvious peak of $Cu^{2+}$ species around 8996 eV in the XANES spectra (Fig. S4)[38,39]. The results further indicate that the redispersion of Cu NPs into Cu single atoms occurs more rapidly in $O_2$-$H_2O$, in accordance with HAADF-STEM and UV-Vis results.

It has been revealed that surface hydroxyl (OH) groups significantly affect the redispersion of metal NPs[14,40]. According to the exchange reaction $D_2 + OH \rightarrow OD + HD$, the HD signal in the H-D exchange experiment can be used to characterize OH content on support surface[41]. As shown in Fig. S5, H-D exchange results demonstrate that there is no significant difference in surface OH content of $\gamma$-$Al_2O_3$ support treated in Ar-$H_2O$ and $O_2$-$H_2O$ atmospheres. Thus, support surface hydroxylation is not the decisive reason for the different redispersion behavior of Cu NPs under Ar-$H_2O$ and $O_2$-$H_2O$ atmospheres.

Quasi in-situ XPS experiments were thus conducted to identify the surface Cu species after treatment in the various atmospheres. Cu $2p_{3/2}$ peak located at 932.4 eV is observed in the fresh 2Cu/AlOOH-900 sample (Fig. 2c), which is assigned to $Cu^+$/$Cu^0$ species[42,43]. The kinetic energy of the main Cu $L_3$VV Auger peak at 916.6 eV and a weak peak around 922.0 eV indicate that $Cu^+$ and a small amount of $Cu^0$ species coexist on the surface of fresh 2Cu/AlOOH-900 sample[43] (Fig. S6). After treatment in $O_2$, Ar-$H_2O$ and $O_2$-$H_2O$ atmospheres for 4 h, the main Cu $2p_{3/2}$ peak shifts to 933.4 eV, characteristic for $Cu^{2+}$ species in CuO, which is further confirmed by Cu $L_3$VV Auger peak at 918.1 eV[43]. While the peak at 932.4 eV indicates that $Cu^+$ species still exist in the samples treated in $O_2$ and Ar-$H_2O$ for 4 h, consistent with Cu $L_3$VV Auger peak at 916.6 eV (Fig. S6)[43]. Interestingly, a Cu $2p_{3/2}$ peak at 935.6 eV and a Cu $L_3$VV Auger peak at 914.4 eV appear in the samples treated in Ar-$H_2O$ and $O_2$-$H_2O$ for 4 h, corresponding to hydroxylated

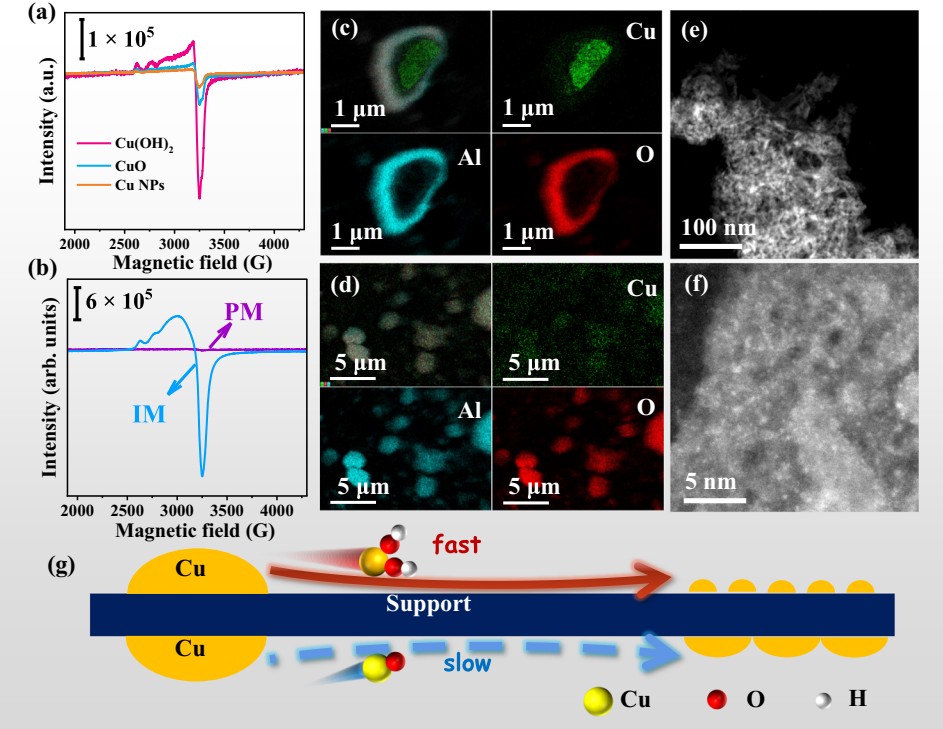

**Fig. 3 | Effect of Cu precursors on redispersion at RT. a** Quasi in-situ EPR spectra of physical mixtures of AlOOH-900 and different Cu precursors in Ar-H$_2$O atmospheres for 24 h. **b** EPR spectra of Cu(OH)$_2$-AlOOH-900 before (PM) and after water immersion (IM) for 24 h. EDX mapping images over Cu(OH)$_2$-AlOOH-900 (**c**) before and (**d**) after water immersion for 24 h. **e, f** HADDF-STEM images of Cu(OH)$_2$-AlOOH-900 after water immersion for 24 h. **g** Scheme of the effect of migration species on the redispersion process.

Cu (Cu-OH) species (Fig. 2c)[43]. As the treatment time extends from 4 to 24 h, the proportion of Cu-OH in Ar-H$_2$O and O$_2$-H$_2$O gradually increases from 4.4% to 7.1% and from 14.3% to 21.7%, respectively. The proportion of Cu-OH species related to total Cu species in the O$_2$-H$_2$O treated sample is much higher than those in the samples treated in Ar-H$_2$O and O$_2$, indicating that Cu-OH species are easily formed in O$_2$-H$_2$O (Fig. 2d). It is worth noting that if Cu NPs are firstly oxidized to CuO in O$_2$ and then exposed to Ar-H$_2$O at RT for 24 h, the proportion of Cu-OH species reaches 10.4%, which is much lower than that of Cu NPs exposed to O$_2$-H$_2$O (21.7%), but higher than Cu NPs directly exposed to Ar-H$_2$O (7.1%) (Fig. S7).

The above results suggest that spontaneous redispersion of Cu NPs in O$_2$-H$_2$O may occur through the oxidation of Cu atoms into atomic Cu-O species, followed by quick hydroxylation into atomic Cu-OH species and final capture by the support. Furthermore, we find that the oxidative redispersion process relies on the size of Cu particles. The redispersion of micron-sized Cu particles is very slow compared with nano-sized Cu particles under the same conditions (Fig. S8). Overall, combining EPR, XAS and UV-Vis results, it can be reasonably inferred that the formation of Cu-OH species is the key to promoting the redispersion of Cu NPs.

## Effect of Cu precursors on the redispersion process

Commercial Cu, CuO, and copper hydroxide (Cu(OH)$_2$) powders with similar particle size were mixed with AlOOH-900, and then treated in Ar-H$_2$O at RT for 24 h to investigate the effect of Cu precursors on the redispersion process. As shown in Fig. 3a, a rather stronger EPR signal around 3300 G is detected using Cu(OH)$_2$ as the precursor as compared to Cu and CuO, implying that Cu(OH)$_2$ is easier to be redispersed. The above results confirm the important role of hydroxylated Cu-OH species in the Cu redispersion process.

Inspired by the crucial role of H$_2$O in the redispersion process, we further investigated whether the redispersion process in gas-phase H$_2$O could be extended to liquid-phase H$_2$O. Cu(OH)$_2$ powder was physically mixed with AlOOH-900 (denoted as Cu(OH)$_2$-AlOOH-900) and immersed in liquid-phase H$_2$O, followed by stirring for 24 h at RT. The obvious diffraction peak at 23.8° and the absence of EPR signal at 3300 G (Fig. S9 and Fig. 3b) in Cu(OH)$_2$-AlOOH-900 sample indicate that only bulk Cu(OH)$_2$ exists in the physical mixture[44]. Large Cu-based particles are also observed in scanning electron microscope and energy-dispersive X-ray spectroscopy (SEM-EDX) mapping images (Fig. 3c). After the water immersion treatment, the diffraction peak of Cu(OH)$_2$ at 23.8° vanishes, and a strong EPR signal at 3300 G appears (Fig. S9 and Fig. 3b). The results indicate that Cu(OH)$_2$ aggregates are redispersed into highly dispersed Cu species which are uniformly distributed on AlOOH-900 surface as shown by the SEM-EDX mapping images (Fig. 3d).

HAADF-STEM images (Fig. 3e, f) further confirm that clusters around 2 nm form after the water immersion. Significantly, EPR signal of the sample treated in liquid-phase H$_2$O is much stronger than that in gas-phase H$_2$O (Fig. 3a, b). The above results reveal that Cu-OH species rather than other Cu intermediates such as CuO$_x$ dominate the fast redispersion process at RT (Fig. 3g), highlighting the indispensable role of H$_2$O in the redispersion of Cu particles at RT. Apart from inducing the formation of Cu-OH species, H$_2$O treatment would also cause the surface hydroxylation of the γ-Al$_2$O$_3$ support. H-D exchange results show about a two-fold increase in the OH content compared to O$_2$ treatment (Fig. S10). Therefore, the impact of surface hydroxylation of the γ-Al$_2$O$_3$ support on metal redispersion may not be ignored and was investigated in detail.

## Role of surface OH groups in the Cu redispersion

For the spontaneous monolayer dispersion phenomenon proposed by Xie et al., active components will disperse onto support surface in the form of atoms or small clusters which cannot be detected by XRD[45]. The amount of dispersed active components as a monolayer is defined

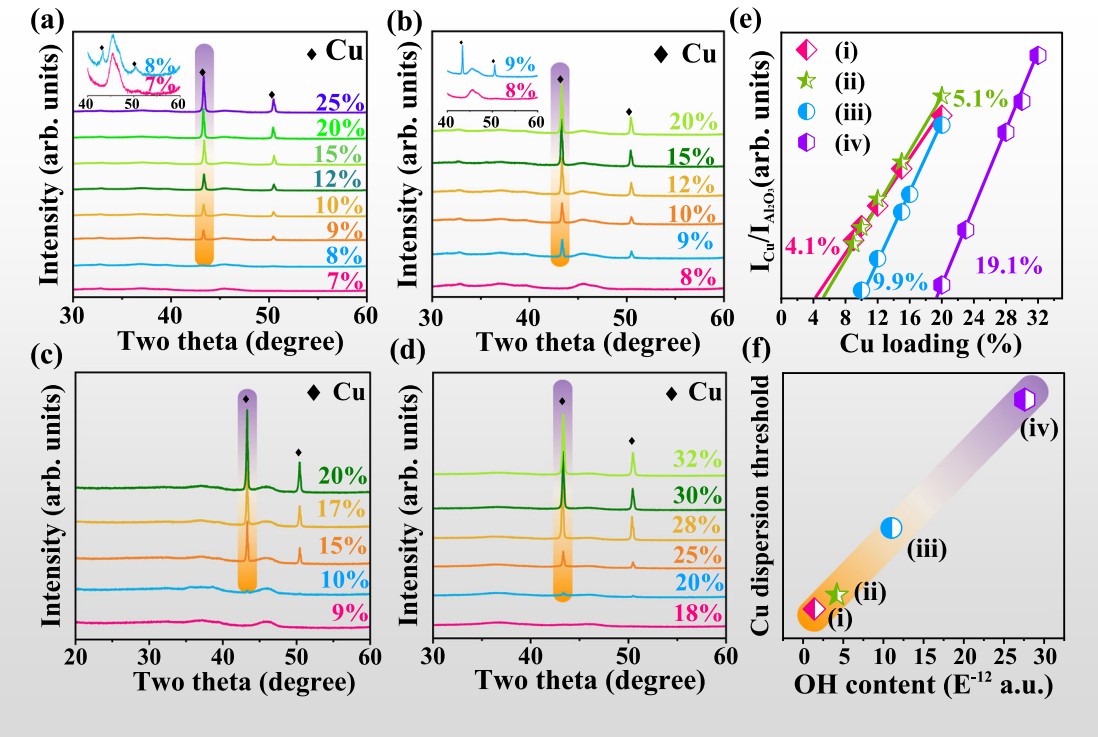

**Fig. 4 | Effect of surface OH content of γ-Al₂O₃ on Cu dispersion threshold at RT for 24 h.** XRD patterns of (**a**) $x$Cu/AlOOH-900 treated in $O_2$ and partial magnification of XRD at 40–60° under 7 and 8 wt.% Cu loading (inset). **b** $x$Cu/AlOOH-900 treated in $O_2$-$H_2O$ and partial magnification of XRD at 40–60° under 8 wt.% Cu loading (inset). **c** $x$Cu/AlOOH-900-$H_2O_2$ treated in $O_2$ and (**d**) $x$Cu/AlOOH-500 treated in $O_2$. **e** Relative intensity of Cu (111) diffraction peak vs. γ-Al₂O₃ (400) diffraction peak for various Cu/γ-Al₂O₃ samples derived from the XRD results. **f** Correlation between OH content and Cu dispersion threshold. (**i**) $x$Cu/AlOOH-900 treated in $O_2$, (**ii**) $x$Cu/AlOOH-900 treated in $O_2$-$H_2O$, (**iii**) $x$Cu/AlOOH-900-$H_2O_2$ treated in $O_2$ and (**iv**) $x$Cu/AlOOH-500 treated in $O_2$.

as the dispersion threshold, above which diffraction peaks of the active components can be observed in XRD[46–48]. The dispersion threshold of an adsorbate on a support can be determined by this well-established method[46]. For supported Cu samples treated in $O_2$ at RT for 24 h (Fig. 4a), no diffraction peaks of Cu species are detectable when Cu loading is below 8 wt.%, confirming that all Cu species are well dispersed on γ-Al₂O₃. Above 8 wt.% Cu loading, additional small peaks at 43.3 and 50.4° are detected which are assigned to metallic Cu (PDF # 04-0836), and intensities of these peaks increase with the increased Cu loadings. Cu (111) peak intensity ($I$Cu) has been normalized by that of γ-Al₂O₃ (400) peak ($I$Al₂O₃) for each sample to obtain $I$Cu/$I$Al₂O₃. From the plot of $I$Cu/$I$Al₂O₃ with the Cu loading, the correlation line intersects the $x$-axis at a point which corresponds to the Cu dispersion threshold at 4.1 wt.% (Fig. 4a, e). The dispersion threshold of $O_2$-$H_2O$ treated sample is found to be 5.1 wt.% using the same method (Fig. 4b, e). The results suggest that the addition of $H_2O$ has effectively increased the dispersion threshold of Cu species on γ-Al₂O₃ surface, which is proposed to result from the enriched surface OH groups as discussed below.

Al₂O₃ supports with different OH contents (Fig. S10) were used to explore the crucial role of surface OH groups in the spontaneous redispersion of Cu NPs. Surface OH content of γ-Al₂O₃ was tuned through treatment in $H_2O_2$ solution or calcination of the pseudo-boehmite precursor at various temperatures[14]. As the OH contents are controlled to be 10 and 20 times higher than that of AlOOH-900 (Fig. S10), the dispersion threshold increases up to 9.9 wt.% and 19.1 wt.% on AlOOH-900-$H_2O_2$ and AlOOH-500 (Fig. 4c–e), respectively. The positive correlation between dispersion threshold and surface OH content (Fig. 4f) suggests that increasing OH content raises the dispersion capacity of γ-Al₂O₃ support by providing more anchoring sites for Cu species.

The role of surface OH on the redispersion of Cu NPs was further demonstrated by using various supports. TEM images in Fig. 5a, b show that Cu NPs with similar sizes around 6 nm exist on h-BN and Si₃N₄ surfaces. After $O_2$-$H_2O$ treatment at RT for 24 h, no particles are observed on the surface of h-BN but a large number of Cu particles still remain on the surface of Si₃N₄ (Fig. 5c, d), which is also consistent with XRD data (Fig. S11). The results indicate that redispersion of Cu NPs can occur on h-BN surface but not on Si₃N₄ surface. Quasi in-situ XPS results show that Cu-OH species can form on the 2Cu/Si₃N₄ and 2Cu/h-BN samples after $O_2$-$H_2O$ treatment (Fig. S12), and thus the different behavior is determined by the supports rather than mobile Cu-OH species. H-D exchange experiments on pure supports indicate that the $O_2$-$H_2O$ treatment significantly increases the OH content on OH-free h-BN surface while no OH groups exist on Si₃N₄ surface even after the $O_2$-$H_2O$ treatment (Fig. 5e). Subsequently, different oxide supports are selected to investigate the effect of surface hydroxylation on the redispersion of Cu NPs, including $SiO_2$, $ZrO_2$, $TiO_2$, $CeO_2$, and zeolites (MCM-41, SSZ-13 and H-beta), which can be hydroxylated (Fig. S13a–g), and nano BN which is not able to be hydroxylated (non-hydroxylated, Fig. S13h). XRD patterns show that Cu NPs can be redispersed on the hydroxylated supports (Fig. S14a–g) rather than the non-hydroxylated ones (Fig. S14h). The above results clearly confirm that surface hydroxylation is essential for the redispersion of Cu NPs.

Based on the above results we infer that the redispersion of Cu NPs under ambient conditions relies on two essential factors: the abundance of hydroxyl (OH) sites on the support surface and the generation of hydroxylated Cu-OH species. For supported Cu samples treated in $O_2$, $CuO_x$ species formed by oxidation in $O_2$ may migrate on the γ-Al₂O₃ surface which however happen slowly at RT. In $H_2O$-containing atmosphere, $H_2O$ significantly increases the

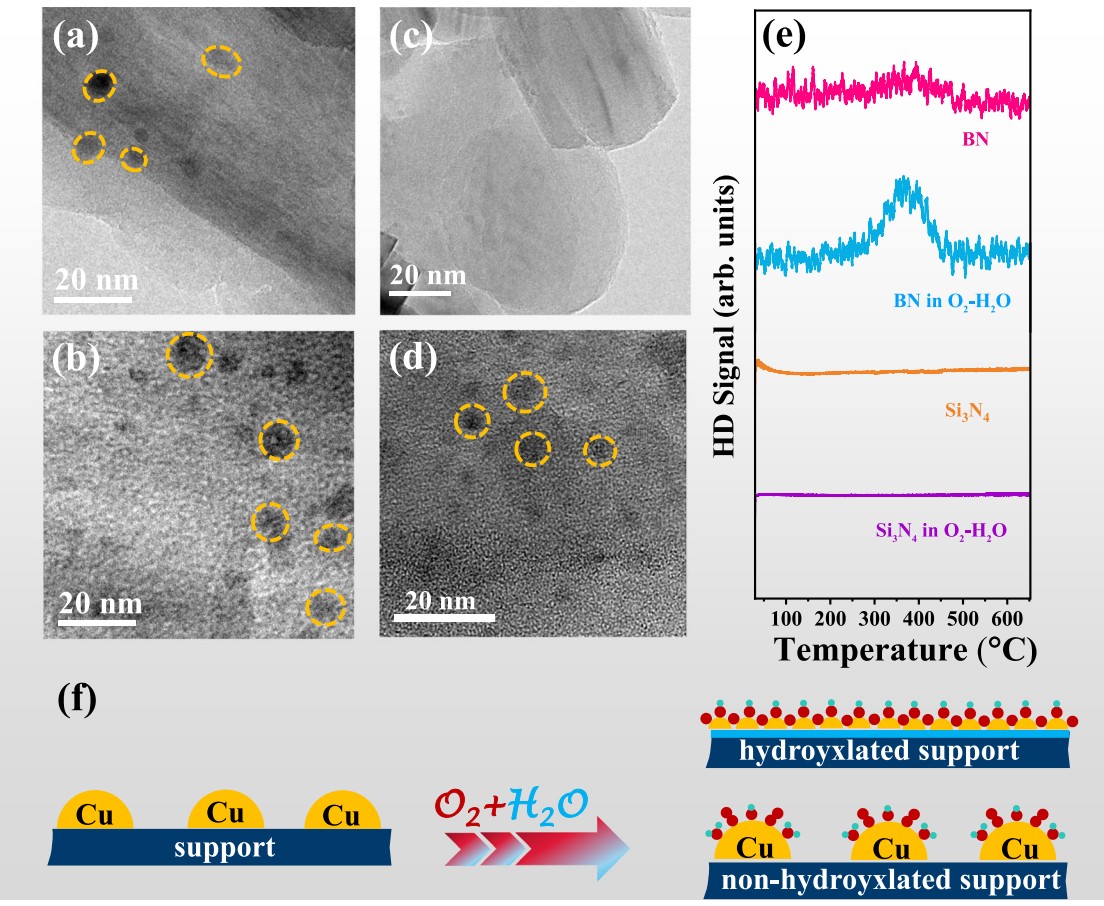

**Fig. 5 | Support effect in the redispersion of Cu NPs at RT.** TEM images of (**a**) 2Cu/BN, (**b**) 2Cu/Si$_3$N$_4$, (**c**) 2Cu/BN catalysts after O$_2$-H$_2$O treatment for 24 h, and (**d**) 2Cu/Si$_3$N$_4$ catalyst after O$_2$-H$_2$O treatment for 24 h. **e** H-D exchange curves of BN and Si$_3$N$_4$ supports after treatment in O$_2$ and O$_2$-H$_2$O atmospheres for 24 h. **f** Scheme of support effect on water-assisted redispersion of Cu NPs at RT.

hydroxylation degree of the γ-Al$_2$O$_3$ support, providing more anchoring sites and migration channels for surface Cu atoms. More crucially, the O$_2$-H$_2$O environment fosters the creation of more mobile hydroxylated Cu-OH species, which expedites the redispersion process (Fig. 3g). Thus, the role of H$_2$O can be interpreted in two distinct manners: first, it promotes the formation of mobile hydroxylated Cu species (in the presence of O$_2$) to speed up the migration of Cu species (kinetic aspect) (Fig. 3g); second, it enriches surface OH groups to provide more anchoring sites (thermodynamic aspect) (Fig. 5f). Consequently, the redispersion of Cu NPs under humid ambient conditions occurs through the synergistic promotion of H$_2$O in both kinetics and thermodynamics.

### Effect of Cu redispersion on catalytic performance of RWGS and CO-PROX reactions

Due to its high CO selectivity and activity, as well as its low cost compared to gold and platinum, copper-based catalysts are one of the most promising candidates for RWGS reaction[49,50]. Fig. S15 shows that both fresh 2Cu/AlOOH-900 as well as the O$_2$-H$_2$O treated sample exhibit 100% selectivity towards CO at 450 °C. The fresh 2Cu/AlOOH-900 catalyst shows a CO$_2$ conversion rate of 5 mmol$_{CO_2}$/g$_{cat}$/h and the CO$_2$ conversion rate is greatly enhanced by up to 7 times on the O$_2$-H$_2$O treated sample (~ 34 mmol$_{CO_2}$/g$_{cat}$/h) (Fig. 6a), which is comparable to the highly dispersed copper catalysts in the literatures[49,50]. Since AlOOH-900 support before and after O$_2$-H$_2$O treatment shows no catalytic activity (Fig. S15), and thus the higher CO$_2$ conversion rates obtained in the O$_2$-H$_2$O treated sample could be attributed to the

better dispersion degree of Cu species. Unfortunately, the reaction rate decreases sharply from 34 to 10 mmol$_{CO_2}$/g$_{cat}$/h for 9 h at 450 °C and the reaction rate decreases in the second reaction cycle (Fig. S15). The obvious diffraction peaks of metallic Cu appear in the XRD spectra (Fig. S16 (i), (ii)), indicating that the highly dispersed Cu species have sintered during the reaction process. After treating the sintered Cu catalyst with O$_2$-H$_2$O at RT, the CO$_2$ conversion rate returns to the initial state (~ 33 mmol$_{CO_2}$/g$_{cat}$/h). Interestingly, the deactivation-activation process can be repeated through cycles of high-temperature reaction and O$_2$-H$_2$O treatment at RT.

Copper-ceria is known as one of the most active catalysts for CO-PROX reaction at the temperature range of 100–140 °C, offering high CO conversion (60–100%) and O$_2$ selectivity (60–100%)[51,52]. Figure 6b shows that O$_2$-H$_2$O treated sample exhibits higher reactivity compared to the fresh 5Cu/CeO$_2$ (~ 35% CO conversion and ~ 47% O$_2$ selectivity), achieving nearly 90% CO conversion and 72% O$_2$ selectivity, demonstrating a competitive advantage compared to the published literatures[51,52]. Nevertheless, CO conversion of the O$_2$-H$_2$O treated sample drops from 89% to 72%, and the selectivity decreases from 74% to 56% after 12 h reaction, suggesting that the catalyst also undergoes Cu sintering in an H$_2$-rich reaction atmosphere (confirmed by XRD in Fig. S16 ((iii), (iv))). Fortunately, the regeneration (both CO conversion and O$_2$ selectivity) can be easily achieved through simple O$_2$-H$_2$O treatment at RT (Fig. 6b, Fig. S17). The reaction results that the Cu-based catalysts undergo sintering and deactivation during the reaction processes, while O$_2$-H$_2$O treatment at RT can serve as an effective reactivation method to regenerate the catalysts.

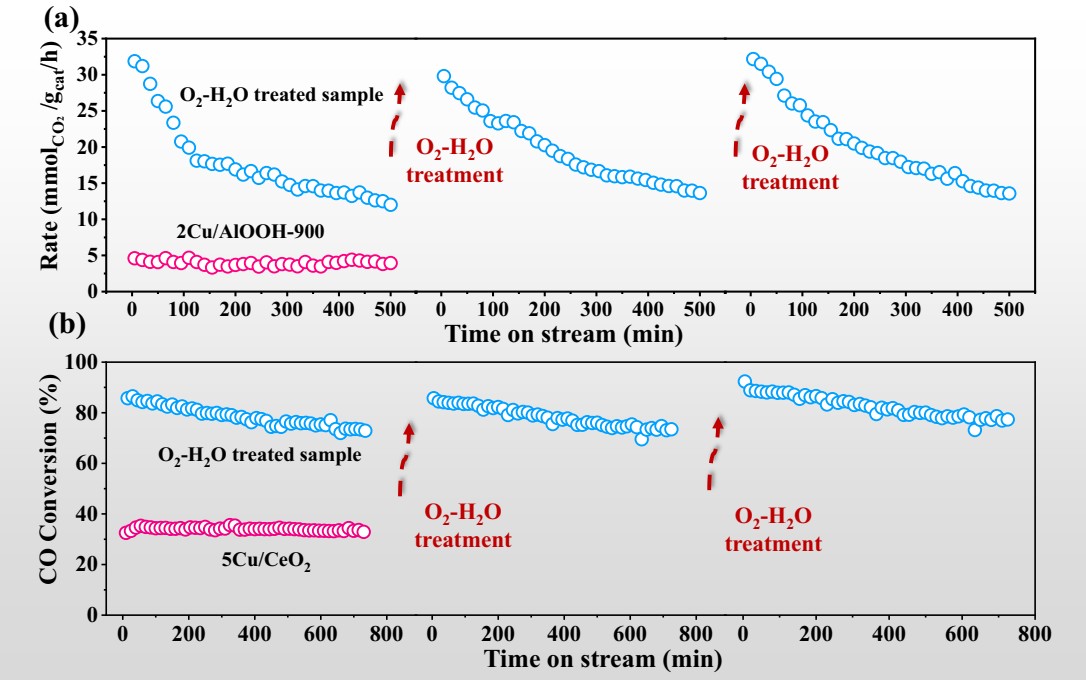

**Fig. 6 | Catalytic performance of Cu-based catalysts.** Stability test of (**a**) 2Cu/AlOOH-900 catalyst before and after $O_2$-$H_2O$ treatment for RWGS reaction; **b** 5Cu/CeO_2 catalyst before and after $O_2$-$H_2O$ treatment for CO-PROX reaction. The deactivation-activation process can be repeated through $O_2$-$H_2O$ treatment at RT.

RWGS reaction condition: 450 °C, weight hourly space velocity (*WHSV*) = 36000 mL/g_cat·h, 24%$CO_2$/72%$H_2$/4%$N_2$, $P$ = 0.1 MPa; CO-PROX reaction condition: 120 °C, *WHSV* = 36000 mL/g_cat·h, 1% CO/0.5% $O_2$/1% $N_2$/97.5% $H_2$, $P$ = 0.1 MPa.

The redispersion of Cu NPs into highly dispersed $Cu^{2+}$ species in $O_2$-$H_2O$ contributes to the enhanced performance and recovered activity for RWGS and CO-PROX reactions. Besides, the redispersion of Cu NPs provides an effective method to tune the size of Cu catalyst, which can be applied in various scenarios, such as Cu NPs for CO oxidation[53] and isolated Cu atoms for selective catalytic reduction of NO$_x$ with $NH_3$ (NH_3-SCR) reaction[54] (Fig. S18).

## Discussion

Spontaneous redispersion of aggregated Cu particles can occur on γ-$Al_2O_3$ surface in the humid environment at RT. The formation of hydroxylated Cu species and enriched surface OH groups under the condition are the two key factors for the facile redispersion of Cu NPs. Hydroxylated Cu species, generated in $O_2$-$H_2O$ atmosphere, act as a mobile species, enabling the copper diffusion to be kinetically feasible at RT. On the other hand, the highly hydroxylated surface induced by $H_2O$ provides numerous anchoring sites of OH for diffusing Cu species, facilitating an energy-favorable configuration of highly dispersed Cu species. The treated Cu catalysts exhibit better performance in the RWGS and CO-PROX reactions. Most notably, sintering of the Cu-based catalysts during reactions can be easily reversed through simple $O_2$-$H_2O$ treatment at RT. This work provides an effective method for the regeneration of sintered Cu catalysts as well as deepens the understanding of the role of both atmosphere and support in the redispersion of metal particles under relatively mild conditions.

## Methods
### Sample preparation

γ-$Al_2O_3$ supports were prepared by calcination of pseudo-boehmite at 500 and/or 900 °C, at a heating rate of 2 °C·min⁻¹ and a dwell time of 5 h (denoted as AlOOH-*T*, in which T represents the calcination temperature)[14]. To increase the surface OH sites, the AlOOH-*T* sample was subject to treatment in hydrogen peroxide ($H_2O_2$) solution at RT for 24 h (AlOOH-*T*-$H_2O_2$)[14]. Commercially available supports, such as

h-BN (98.5% purity, 1 μm, Shanghai Aladdin Biochemistry technology Co., Ltd.), Nano BN (99.9% purity, <200 nm, Shanghai Macklin biochemical Co., Ltd.), $ZrO_2$ (99.9% purity, ≤ 100 nm, Shanghai Aladdin Biochemistry technology Co., Ltd.), $TiO_2$ (99.8% purity, rutile, 40 nm, Shanghai Aladdin Biochemistry technology Co., Ltd.), $Si_3N_4$ (99.3% purity, 325 mesh, Alfa Aesar, USA.), $CeO_2$ (99.9% purity, 50 nm, Shanghai Macklin biochemical Co., Ltd.), SSZ-13 (3–10 μm, Dalian Ze'er Catalytic Materials Co., Ltd.), H-beta (2–5 μm powder, Si/Al = 25–30, Dalian Ze'er Catalytic Materials Co., Ltd.), MCM-41 (3–6 μm powder, Pure Silica, Dalian Ze'er Catalytic Materials Co., Ltd.), were used directly. Copper hydroxide (Cu(OH)_2) was purchased from Macklin (99.9% purity, ~ 40 nm) and directly used for the synthesis. 2Cu/AlOOH-900 catalyst was synthesized via conventional wet impregnation. Briefly, AlOOH-900 (2 g) was impregnated with an aqueous solution of Cu(NO_3)_2·3$H_2O$ (1.56 mL, 0.4 mol/L), corresponding to a copper weight loading of 2 wt.%. The resulting sample was dried overnight and calcined at 500 °C for 4 h in air, followed by reduction in pure $H_2$ at 500 °C for 2 h with a flow rate of 100 mL/min. Cu-based catalysts with different Cu weight loadings supported on different supports were prepared according to the above-mentioned methods, and an equal proportion of Cu(NO_3)_2·3$H_2O$ solution was added to 2 g of the supports (for example, 10Cu/AlOOH-900 need to add 7.81 mL Cu(NO_3)_2·3$H_2O$ solution). After reduction in $H_2$ at 500 °C for 2 h, the obtained sample was denoted as *x*Cu/support, where *x* represents the weight loading of Cu. The reduced samples were further treated in different gases such as $O_2$, Ar, and $H_2O$ with a flow rate of 100 mL/min at RT for 24 h. AlOOH-900 (2 g) and Cu(OH)_2 powders (15 mg, Cu weight loading 0.5 wt.%) were physically mixed, and the obtained Cu(OH)_2-AlOOH-900 sample was labeled as PM (Physical mixture). Cu(OH)_2-AlOOH-900 sample was then vigorously stirred in 50 ml water for 24 h at RT before being evaporated at 100 °C to remove the water, and the obtained sample was labeled as IM (Impregnation sample). Water vapor was introduced by passing the gas flow through a bubbler with about 3 vol.% $H_2O$ at RT.

## Sample Characterizations

X-ray diffraction (XRD) patterns were acquired using an Empyrean-100 diffractometer equipped with a Cu Kα radiation source (λ = 1.5418 Å) at 40 kV and 40 mA. Quasi in-situ electron paramagnetic resonance (EPR) spectra were collected at 110 K using a Bruker A200 EPR spectrometer. Ultraviolet-visible (UV-Vis) spectra were acquired in Lambda 950 (Perkin Elmer) equipped with an in-situ reaction cell, in normal or time-dependent modes. Catalyst powders (30–50 mg) were pressed into self-supporting wafers and placed within a temperature-controlled stainless-steel cell equipped with $CaF_2$ windows and connected to a gas manifold. The sample was pretreated at 500 °C for 2 h under a $H_2$ atmosphere with a flow rate of 100 mL/min to obtain the first spectrum, and then the corresponding gas was introduced for 30 min before collection of the next spectrum. High-resolution transmission electron microscopy (HR-TEM) images were recorded on a JEOL JEM 2100 TEM instrument with a 200 kV accelerating voltage. High-angle annular dark-field scanning transmission electron microscopy (HAADF-STEM) images were acquired using a field emission TEM (JEM-F200) with a 200 kV operating voltage. HAADF-STEM images were obtained on a JEM ARM 300 F with a 300 kV accelerating voltage.

Quasi in-situ X-ray photoelectron spectroscopy (XPS) measurements were carried out with a spectrometer equipped with an Mg Kα X-ray source operated at 300 W. Samples were treated in a reaction chamber at ambient pressure and then transferred to the analysis chamber immediately without exposure to air. All XPS binding energy peaks were calibrated by C 1s at 284.6 eV. Quasi in-situ X-ray absorption spectroscopy (XAS) spectra were collected in fluorescence mode at RT at the BL11B beamline of Shanghai Synchrotron Radiation Facility (SSRF). Before measurement, the sample treated for different times was sealed in capillary tubes in the glove box without exposure to air.

H-D exchange experiments were conducted in a homemade microreactor connected with a mass spectrometry (OMNI Star TM). Typically, 0.1 g sample was loaded in a quartz tube and then pretreated under Ar atmosphere at 200 °C for 2 h. After the pretreatment, the H-D exchange experiment was started with a heating rate of 10°/min from RT to 750 °C by recording the mass spectroscopy HD signal ($m/z = 3$).

## Catalytic Test

Reverse water gas shift (RWGS) reactions were tested using a home-made fixed-bed micro-reactor with the weight hourly space velocity (WHSV) of 36,000 mL/$g_{cat}$·h. The 50 mg pelleting catalysts (20 ~ 40 mesh) were loaded in a quartz tube with an inner diameter of 6 mm. The reaction gas contains 24% $CO_2$, 72% $H_2$ (volume ratio), balanced with $N_2$, which was used as an internal standard. Before measurement, the fresh 2Cu/AlOOH-900 catalyst was pretreated by $H_2$ at 500 °C for 1 h to reduce the oxidized Cu species on the sample surface. The $O_2$-$H_2O$ treated sample was directly used for RWGS reaction test. The products were analyzed by an online gas chromatograph (Agilent 490 Micro GC) equipped with a 5 Å molecular sieve column and a thermal conductivity detector. The reaction took place at atmospheric pressure and 450 °C. Only CO is generated during the reaction, and $CO_2$ conversion and CO selectivity were calculated according to the following equations:

$$CO_2 \, Conversion(\%) = \frac{C_{CO_2(inlet)} - C_{CO_2(outlet)}}{C_{CO_2(inlet)}} \times 100\% \quad (1)$$

$$CO \, Selectivity(\%) = \frac{C_{CO(outlet)}}{C_{CO(outlet)} + C_{CH_4(outlet)}} \times 100\% \quad (2)$$

where the subscripts "inlet" and "outlet" are related to the inlet and outlet gas concentrations, respectively.

Preferential oxidation of CO in excess of $H_2$ (CO-PROX) reactivity was tested in a fixed-bed reactor by using 50 mg of sieved catalyst (20–40 mesh) in a gas mixture of 1% CO/0.5% $O_2$/1% $N_2$/97.5% $H_2$ (volume ratio) with a WHSV of 36, 000 mL/$g_{cat}$·h. Prior to the measurement, the fresh sample was pretreated in pure $H_2$ at 500 °C for 1 h. The reaction was performed at 120 °C and the products were analyzed by an online gas chromatograph (Agilent 490 Micro GC) equipped with a 5 Å molecular sieve column and a thermal conductivity detector. The CO conversion and $O_2$ selectivity were calculated according to the following equations:

$$CO \, Conversion(\%) = \frac{C_{CO(inlet)} - C_{CO(outlet)}}{C_{CO(inlet)}} \times 100\% \quad (3)$$

$$O_2 \, Selectivity(\%) = \frac{0.5 \times (C_{CO(inlet)} - C_{CO(outlet)})}{(C_{O_2(inlet)} - C_{O_2(outlet)})} \times 100\% \quad (4)$$

where the subscripts "inlet" and "outlet" are related to the inlet and outlet gas concentrations, respectively.

CO oxidation reaction was tested in a fixed-bed reactor at atmospheric pressure with WHSV of 40,000 mL/$g_{cat}$·h. The reaction gas contains 1% CO, 20% $O_2$, 1% $N_2$, and 78% He. The products were analyzed by an online gas chromatograph (Agilent 490 Micro GC) equipped with a 5 Å molecular sieve column and a thermal conductivity detector.

$$CO \, Conversion(\%) = \frac{C_{CO(inlet)} - C_{CO(outlet)}}{C_{CO(inlet)}} \times 100\% \quad (5)$$

where the subscripts "inlet" and "outlet" are related to the inlet and outlet gas concentrations, respectively.

Selective catalytic reduction of $NO_x$ with $NH_3$ ($NH_3$-SCR) activity tests of the sieved powder catalysts were carried out in a fixed-bed quartz flow reactor at atmospheric pressure with WHSV of 120,000 mL/$g_{cat}$·h. The reaction gas contains 500 ppm NO, 500 ppm $NH_3$, 4 % $O_2$, and balance $N_2$. The effluent gases including NO, $NH_3$, $NO_2$, and $N_2O$ were continuously analyzed by an online Nicolet iS50 IR spectrometer (Nicolet, USA) equipped with a gas cell. The activity data were recorded at every target temperature after stabilizing for 60 min. Finally, $NO_x$ conversion was calculated using the below equations.

$$NO_x \, Conversion(\%) = \frac{C_{NO_x(inlet)} - C_{NO_x(outlet)}}{C_{NO_x(inlet)}} \times 100\% \quad (6)$$

where the subscripts "in" and "out" are related to the inlet and outlet gas concentrations, respectively.

## Data availability

All data that support the findings in this paper are available within the article and its Supporting Information or are available from the corresponding authors upon reasonable request. Source data are provided with this paper.

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

## Acknowledgements

This work was financially supported by National Key R&D Program of China (2021YFA1502800, 2022YFA1504800 and 2022YFA1504500), National Natural Science Foundation of China (No. 21825203, No. 22288201, No. 22332006 and No. 22321002), the Fundamental Research Funds for the Central Universities (20720220009), and Photon Science Center for Carbon Neutrality. R.L. thanks support from the China Postdoctoral Science Foundation (2023M743425). B.W. acknowledges the support from National Natural Science Foundation of China (22302125).

## Author contributions

Y.F. conducted the experiments and drafted the manuscript. R.L. conceived the research and revised the manuscript. B.W. was responsible for the interpretation of EXAFS results. X.F. conducted the SEM and EDX mapping tests. X.D. and C.X.L. helped with the sample pretreatment and EXAFS characterizations. F.W. measured the NH₃-SCR catalytic reactivity. C.H.L and C.D. participated in data analysis. R.M. and Y.N. participated in revising the manuscript. Q.F. directed the project, conceived the research, and revised the manuscript. All the authors discussed the results and participated in writing the manuscript.

## Competing interests

The authors declare no competing interests.
