## [Peer Review File · Nature Communications]

Water-assisted oxidative redispersion of Cu particles through formation of Cu hydroxide at room temperatureReviewers' comments:

Reviewer #1 (Remarks to the Author):

This manuscript reports that Cu metal nanoparticles (NPs) on Al₂O₃ undergo assisted oxidative dispersion into isolated Cu(II) species under H₂O/O₂ at room temperature through formation of CuO and then Cu hydroxide. The catalytic activity of the materials with and without the Cu dispersion is compared for CO oxidation and NH₃-SCR. From viewpoints of heterogeneous catalysis and nanomaterial chemistry, the present work is not among the top level considering the following reasons.

1. It is well known that Cu/Al₂O₃ is not effective catalyst for CO oxidation and NH₃-SCR. So, the catalytic impact of this report is low.

2. Oxidation of small Cu metal NPs to CuO NPs by air at room temperature is a common sense. Auto dispersion of CuO into atomic Cu(II) via Cu(OH)₂ species is a new finding in this work. So, the in situ characterization should be focused on the pathway of CuO+H₂O to Cu(OH)₂ to Cu(II) reaction. For example, EXAFS results should include the data for Cu(0)+O₂, Cu(0)+H₂O, and CuO+H₂O.

3. In situ time resolved EXAFS data is not shown to support the in situ UV-vis results in Fig. 2a.

Knowing that this journal accepts only innovative research, I do not consider the manuscript acceptable for publication in this journal.

Reviewer #2 (Remarks to the Author):

This work reports an efficient strategy to disperse Cu particles into single atoms or ultrasmall clusters under water-assisted oxidation treatment. Through investigating such a method on various substrates including γ -Al₂O₃, ZrO₂, TiO₂, MgO, h-BN, Si₃N₄, and flake graphite, and combining a variety of different characterizations they concluded that the critical role of H₂O is promoting the formation of mobile hydroxylated Cu species and simultaneously providing enriched anchoring sites for the single atomic Cu species. Moreover, the profit of such a particle-size tunability was clearly demonstrated in two model reactions. I consider this is an important work that can provoke the broad interests in the heterogeneous community. The paper has been well organized and well written. Therefore, I would be happy to recommend its publishing on the journal of Nature Communications. A minor revision may be needed according to the following concerns:

1. There seems to be a clerical error in line 120, Fig. 1e should be Fig 1f. Also, there were no claims of the corresponding lengths for each scale bar in the HAADF-STEM images as shown in Fig. 1.

2. In line 132-135, the XPS observed the formation of Cu²⁺ but cannot discern the Cu⁺ and Cu⁰ on the O₂- and Ar-H₂O treated samples. Then the AES was further applied to exclude the existence no Cu⁰ species. Could the authors please give a more detailed explanation? Why no Cu²⁺ species was recognized on the AES spectra?

3. In line 255-256, the conclusion of “The Cu particle size may be the decisive factor for the different catalytic performance in the two reactions” may be a bit overstated. This is because in this specific study, the oxidation state of the Cu species is closely related to the particle size. However, this may not always be true for other recipe of catalyst preparations. In other words, the valence state of the Cu species may also contribute important roles in these reactions.

Reviewer #3 (Remarks to the Author):

Fan et al. report on the effect of wet pretreatments of supported Cu nanoparticles on metal dispersion. While some interesting data are produced, I am not sure that the article meets the standards of Nature Communications.

1. The introduction does not summarize the current knowledge on metal redispersion. It is more focused on a few examples dealing with Cu.
2. The main oxide supports, referred to as AlOOH-500 and -900, are claimed to be γ -Al₂O₃ (page 4), but there is no evidence for that phase.
3. Electron microscopy images in Figs. 1 and 4 show scale bars with no indication of the scale. In Fig. 1a, the supposed single Cu atoms are surrounded by circles that make any visualization impossible.
4. In Fig. 1f, EXAFS data are reported. However, neither XANES curves nor EXAFS fitting is provided. The components at R=4-5 Å are not ascribed for the Cu foil.
5. Moreover, still for EXAFS, why only 2Cu/AlOOH-900 sample treated in O₂-H₂O is reported? One would at least expect the data for this sample before treatment.
6. The XPS data in Fig. 2b are hardly visible, hidden by thick fitting curves. Moreover, why are the 2p_{1/2} components not shown?
7. H-D exchange experiments: describe them and how they can be interpreted in terms of OH content, since nothing can be understood just from the claim “H-D exchange results show that Ar-H₂O treatment increases the OH content...”.
8. Page 8, what is the “spontaneous monolayer dispersion phenomenon”? The way it is described, it looks more like coalescence than dispersion.
9. By the way, the use of the “dispersion” term throughout the paper is often improper. “Dispersion” refers to the fraction of surface metal atoms in a nanoparticle. The process leading to single atoms or clusters from bigger entities (such as nanoparticles) is called “redispersion”.
10. Page 11, it is mentioned that XPS shows that hydroxylated Cu species “can” form on Si₃N₄, but H-D exchange indicates that few OH groups are present on Si₃N₄. However, it is concluded that surface hydroxylation is key to Cu redispersion. How does it account for XPS results?
11. Catalytic data (Fig. 6) look minimalist. For example, I would expect several successive heating/cooling cycles to assess the stability of the process. Would the two samples keep exhibiting different behaviors?
12. The experimental data are only partial. For instance, plenty of parameters such as reactor diameter, flow rates, catalyst weight, etc., are omitted. Isn't the conversion formula (page 17) valid for CO oxidation?

13. Fig S8d: the flat background is surprising.

14. Check “corporation” (page 3) and “conductive” (page 13) terms.

Overall, due to the lack of important information as well as the poor analysis and presentation of the data, I do not recommend this paper for publication, at least in its present form.

Point-by-Point Response to the Comments

Reviewer #1: This manuscript reports that Cu metal nanoparticles (NPs) on Al₂O₃ undergo assisted oxidative dispersion into isolated Cu(II) species under H₂O/O₂ at room temperature through formation of CuO and then Cu hydroxide. The catalytic activity of the materials with and without the Cu dispersion is compared for CO oxidation and NH₃-SCR. From viewpoints of heterogeneous catalysis and nanomaterial chemistry, the present work is not among the top level considering the following reasons.

Response: We really appreciate the referee for carefully reading the manuscript and for making insightful suggestions to help us significantly improve the quality of our manuscript. We have performed additional experiments including catalytic activity tests (reverse water gas shift (RWGS) and preferential oxidation of carbon monoxide (CO-PROX) reactions) and more quasi *in-situ* characterizations to strengthen the catalytic impact and to reveal the underlying mechanisms. Based on the experiment results, we revised our manuscript carefully, and sincerely hope that our revisions have satisfactorily addressed the reviewer's concerns. A new version is attached for review. The following is the point-by-point response:

1. It is well known that Cu/Al₂O₃ is not effective catalyst for CO oxidation and NH₃-SCR. So, the catalytic impact of this report is low.

Response: We appreciate the constructive comments raised by the referee. Cu/Al₂O₃ catalyst has been demonstrated to be an active catalyst for reverse water gas shift (RWGS) reaction (*ACS Catal.*, 2019, 9, 6243-6251; *Chem. Commun.*, 2021, 57, 1153-1156). As shown in **Fig. R1a**, O₂-H₂O treated sample shows a significantly enhanced CO₂ conversion rate (~ 34 mmol_{CO₂}/g_{cat}/h) by up to about 7 times compared to the 2Cu/AlOOH-900 (~ 5 mmol_{CO₂}/g_{cat}/h). In addition, copper oxide clusters supported on ceria are proved to be one of the most active catalysts for preferential oxidation of carbon monoxide (CO-PROX) reaction (*ACS Catal.*, 2015, 5, 2088-2099; *Appl. Surf. Sci.*, 2022, 600, 154100). **Fig. R1b** shows that about double CO conversion is observed after O₂-H₂O treatment of 5Cu/CeO₂. The reaction rate of the O₂-H₂O treated sample in the two reactions is comparable to other highly dispersed Cu catalysts at similar conditions in the literatures (*ACS Catal.*, 2019, 9, 6243-6251; *Chem. Commun.*, 2021, 57, 1153-1156; *ACS Catal.*, 2015, 5, 2088-2099; *Appl.*

Surf. Sci., 2022, 600, 154100), indicating that O₂-H₂O treatment is powerful to generate highly dispersed Cu species with high activity in RWGS and CO-PROX reactions.

Fig. R1 Catalytic performance of Cu-based catalysts. Stability test of (a) 2Cu/AlOOH-900 catalyst before and after O₂-H₂O treatment for RWGS reaction; (b) 5Cu/CeO₂ catalyst before and after O₂-H₂O treatment for CO-PROX reaction. The deactivation-activation process can be repeated through O₂-H₂O treatment at RT. RWGS reaction condition: 450 °C, weight hourly space velocity (WHSV) = 36000 mL/g_{cat}·h, 24% CO₂/72% H₂/4% N₂, *P* = 0.1 MPa; CO-PROX reaction condition: 120 °C, WHSV = 36000 mL/g_{cat}·h, 1% CO/0.5% O₂/1% N₂/97.5% H₂, *P* = 0.1 MPa. This is used as **Fig. 6** in the revised manuscript.

Besides high catalytic activity, the regeneration of a sintered catalyst is also crucial, drawing a lot of attentions for decades. It should be noted that the deactivated Cu/Al₂O₃ and Cu/CeO₂ catalysts during high temperature reactions (RWGS and CO-PROX) can be easily reactivated by exposing to O₂-H₂O atmosphere at RT (**Fig. R1**). Compared to other regeneration strategy which commonly requires high temperature and specific gases (such as O₂, NH₃ and CH₃I) (*Science*, 2016, 353, 150-154; *J. Am. Chem. Soc.*, 2019, 141, 4505–4509; *ACS Catal.*, 2012, 2, 552–560), such a simple regeneration method under the ambient condition is environmentally friendly and energy saving.

The new catalytic performance experiments have been added as **Fig. 6** with

corresponding discussions on **Pages 16 - 18** in the revised manuscript: “**Effect of Cu redispersion on catalytic performance of RWGS and CO-PROX reactions.** Due to the high CO selectivity and activity of copper, as well as its low cost compared to gold and platinum, copper-based catalysts may be one of the most promising candidates for RWGS reaction^{50,51}.”

2. Oxidation of small Cu metal NPs to CuO NPs by air at room temperature is a common sense. Auto dispersion of CuO into atomic Cu(II) via Cu(OH)₂ species is a new finding in this work. So, the *in-situ* characterization should be focused on the pathway of CuO+H₂O to Cu(OH)₂ to Cu(II) reaction. For example, EXAFS results should include the data for Cu(0)+O₂, Cu(0)+H₂O, and CuO+H₂O.

Response: We greatly appreciate the reviewer's valuable suggestion. We agree with the referee that we should focus on the formation pathway of hydroxylated Cu-OH species. We have conducted quasi *in-situ* EXAFS experiments of Cu NPs under different atmospheres. It is difficult to distinguish the highly dispersed CuO_x and Cu(OH)_x species due to the same Cu-O bond distance according to the standard spectra of CuO, Cu₂O and Cu(OH)₂ samples shown in **Fig. S5** of the revised SI. Thus, quasi *in-situ* XPS was used to identify the evolution of Cu(0) or CuO species treated in O₂, H₂O and O₂-H₂O. As shown in **Fig. R2**, no Cu-OH species can be found for Cu NPs in O₂ and only a small proportion of Cu-OH species are observed in Ar-H₂O. In contrast, a large proportion of Cu-OH species are detected in O₂-H₂O. If the Cu NPs are firstly oxidized into CuO in O₂ and then exposed to Ar-H₂O, the proportion of Cu-OH species (10.4%) is much less than Cu NPs in O₂-H₂O (21.7%) but more than Cu NPs directly exposed to Ar-H₂O (7.1%). The result suggests that spontaneous redispersion of Cu NPs in O₂-H₂O may occur through the oxidation of Cu atoms into atomic Cu-O species followed by the hydroxylation of Cu-O species into Cu-OH species, which migrate and get captured by the support. A new cycle is then repeated until the redispersion process is completed.

Fig. R2 Quasi *in-situ* Cu 2p XPS spectra of 2Cu/AlOOH-900 after treatment in different atmospheres. The three images on the left are used as **Fig.2c** in the revised manuscript.

The above XPS results have been added as **Fig. 2c** and **Figs. S7** in the revised manuscript and SI and related discussions are given on **Pages 8, 9** in the revised manuscript: “Quasi *in-situ* XPS experiments were conducted to identify the surface Cu species after treatment in various atmospheres. Cu 2p_{3/2} peak located at 932.4 eV is observed in the fresh 2Cu/AlOOH-900 sample (Fig. 2c), which is assigned to Cu⁺/Cu⁰ species^{42,43}. The kinetic energy of the main Cu L₃VV Auger peak at 916.6 eV and a weak peak around 922 eV indicate that Cu⁺ and a small amount of Cu⁰ species coexist on the surface of fresh 2Cu/AlOOH-900 sample⁴³ (Fig. S6).”

What’s more, controlled experiments on the effect of Cu precursors (Cu, CuO and Cu(OH)₂ with similar particle size) are conducted. As shown in **Fig. R3a**, much more highly dispersed Cu species are observed (stronger EPR signal) using Cu(OH)₂ as precursor in Ar-H₂O. More obvious redispersion can be found for Cu(OH)₂ precursor in liquid-phase H₂O (**Fig. R3b**) than gas-phase H₂O (Ar-H₂O) (**Fig. R3a**), indicating that increasing H₂O amount can significantly promote the redispersion of Cu species. **Figs. R3c-f** show that bulk Cu(OH)₂ can be redispersed into Cu clusters and single atoms on Al₂O₃ surface in liquid-phase H₂O. The above results further demonstrate that the formation of mobile Cu(OH)₂ accounts for the rapid redispersion of Cu species (**Fig. R3g**).

Fig. R3 Cu precursors effect on dispersion of Cu NPs at RT. (a) Quasi *in-situ* EPR spectra of physical mixtures of AlOOH-900 and different Cu precursors in Ar-H₂O atmospheres for 24 h. (b) EPR spectra of Cu(OH)₂-AlOOH-900 before and after water immersion for 24 h. EDX mapping images over Cu(OH)₂-AlOOH-900 (c) before and (d) after water immersion for 24 h. (e, f) HADDF-STEM images of Cu(OH)₂-AlOOH-900 after water immersion for 24 h. Scale bars are 100 nm and 5 nm in (e) and (f), respectively. (g) Scheme of the effect of migration species on the dispersion of Cu NPs. This is used as **Fig. 3** in the revised manuscript.

The above results have been added as **Fig. 3** and related discussions are presented on **Pages 9 - 11** in the revised manuscript: “Cu-OH species dominated the redispersion process. Commercial Cu, CuO, and copper hydroxide (Cu(OH)₂) powders with similar particle size have been mixed with AlOOH-900, and then treated in Ar-H₂O at RT for 24 h to investigate the effect of Cu precursors on the redispersion process.”

3. *In-situ* time resolved EXAFS data is not shown to support the *in-situ* UV-vis results in Fig. 2a.

Response: We thank the referee for the helpful advice. Quasi *in-situ* EXAFS spectra with

different treatment times are shown in **Fig. R4**. The disappearance of Cu-Cu bond (peak at ~ 2.2 Å) and formation of Cu-O bond (peak at ~ 1.5 Å) is much faster in O₂-H₂O (complete redispersion at 4 h) than the cases in O₂ and Ar-H₂O (incomplete redispersion at 4 h and even 24 h), which is consistent with the UV-Vis results. The EXAFS results have been added as **Fig. 2b** and detailed discussions are given on **Pages 7, 8** in the revised manuscript: “Subsequently, quasi *in-situ* XAS experiments were conducted to identify the chemical state of Cu in O₂, Ar-H₂O and O₂-H₂O atmospheres.”

Fig. R4 Quasi *in-situ* Fourier-transforms of k^3 -weighted Cu K-edge EXAFS spectra of 2Cu/AlOOH-900 treated in O₂, Ar-H₂O and O₂-H₂O for 4 h and 8 h, as well as standard samples of Cu foil and CuO. This is used as **Fig. 2b** in the revised manuscript.

Reviewer #2: This work reports an efficient strategy to disperse Cu particles into single atoms or ultrasmall clusters under water-assisted oxidation treatment. Through investigating such a method on various substrates including γ -Al₂O₃, ZrO₂, TiO₂, MgO, h-BN, Si₃N₄, and flake graphite, and combining a variety of different characterizations they concluded that the critical role of H₂O is promoting the formation of mobile hydroxylated Cu species and simultaneously providing enriched anchoring sites for the single atomic Cu species. Moreover, the profit of such a particle-size tunability was clearly demonstrated in two model reactions. I consider this is an important work that can provoke the broad interests in the heterogeneous community. The paper has been well organized and well written. Therefore, I would be happy to recommend its publishing on the journal of Nature Communications. A minor revision may be needed according to the following concerns:

Response: We appreciate very much the referee’s positive comments and valuable

suggestions on our work. We have carefully considered all suggestions and performed additional experiments. A new version is attached for the further review. The following is the point-by-point response:

1. There seems to be a clerical error in line 120, Fig. 1e should be Fig 1f. Also, there were no claims of the corresponding lengths for each scale bar in the HAADF-STEM images as shown in Fig. 1.

Response: Thank the referee very much for pointing out the errors in our manuscript. We have corrected the clerical error and added the missing claims for scale bar in HAADF-STEM images. Corresponding modifications have been made in the revised manuscript.

2. In line 132-135, the XPS observed the formation of Cu^{2+} but cannot discern the Cu^+ and Cu^0 on the O_2 and Ar- H_2O treated samples. Then the AES was further applied to exclude the existence no Cu^0 species. Could the authors please give a more detailed explanation? Why no Cu^{2+} species was recognized on the AES spectra?

Response: Thank the referee so much for the professional suggestions. To monitor the evolution of Cu species in different atmospheres, quasi *in-situ* XPS tests with different treatment times are re-conducted. Commonly, Cu $2p_{3/2}$ peak can be used to identify Cu^{2+} (BE: $\sim 933.6 \text{ eV} \pm 0.2 \text{ eV}$) and Cu^+/Cu^0 (BE: $\sim 932.4 \text{ eV} \pm 0.2 \text{ eV}$), and Cu L_3VV Auger peak is applied to distinguish Cu^+ (KE: $\sim 916.4 \text{ eV} \pm 0.2 \text{ eV}$) and Cu^0 (KE: $\sim 918.6 \text{ eV} \pm 0.2 \text{ eV}$), of which the latter is close to Cu^{2+} ($\sim 918.1 \text{ eV} \pm 0.2 \text{ eV}$).

Fig. R5 Cu $2p$ XPS spectra and Cu L_3VV Auger spectra of Cu, Cu_2O and CuO. (From

Fig. R6 Quasi *in-situ* Cu L₃VV spectra of 2Cu/AlOOH-900 after treatment in O₂, Ar-H₂O and O₂-H₂O atmospheres for 4 h. This is used as Fig. S6 in the new version.

For 2Cu/AlOOH-900 treated in O₂, Ar-H₂O and O₂-H₂O, the much broader Cu 2*p* and Cu L₃VV Auger peaks compared with standard Cu₂O and CuO samples (Fig. R5, Surf. Interface Anal., 2017, 49, 1325–1334) indicate that multiple Cu species exist. According to the position of Cu L₃VV Auger peak and deconvolution of Cu 2*p* peaks (Figs. R2, 6), the Cu species in these samples can be exactly recognized: (1) Cu⁺ and Cu²⁺ in O₂ as proved by the Cu 2*p*_{3/2} peaks at 932.4 and 933.4 eV and Cu L₃VV Auger peak at 916.4 and 918.1 eV; (2) Cu⁺, Cu²⁺ and Cu-OH in Ar-H₂O and O₂-H₂O as confirmed by the Cu 2*p*_{3/2} peaks at 932.4, 933.4 eV and 935.6 eV with corresponding Cu L₃VV Auger peak at 916.4, 918.1 and 914.4 eV, respectively. We exclude the existence of Cu⁰ species in Ar-H₂O and O₂-H₂O treated samples based on the AES results (Figs. R5, 6).

The Cu 2*p* and Cu L₃VV spectra are updated as Fig. 2c and Fig. S6 in the revised manuscript and SI with corresponding discussions on Pages 8, 9 in the revised manuscript: “Quasi *in-situ* XPS experiments were conducted to identify the surface Cu species after treatment in various atmospheres. Cu 2*p*_{3/2} peak located at 932.4 eV is observed in the fresh 2Cu/AlOOH-900 sample (Fig. 2c), which is assigned to Cu⁺/Cu⁰ species^{42,43}. The kinetic energy of the main Cu L₃VV Auger peak at 916.6 eV and a weak peak around 922 eV

indicate that Cu⁺ and a small amount of Cu⁰ coexist in the surface of fresh 2Cu/AlOOH-900 sample⁴³ (Fig. S6).”

3. In line 255-256, the conclusion of “The Cu particle size may be the decisive factor for the different catalytic performance in the two reactions” may be a bit overstated. This is because in this specific study, the oxidation state of the Cu species is closely related to the particle size. However, this may not always be true for other recipe of catalyst preparations. In other words, the valence state of the Cu species may also contribute important roles in these reactions.

Response: Thank the referee so much for the constructive suggestion. We fully agree with the referee that both the valence state and the size of Cu species play important roles in catalytic reactions. In our studies, the enhanced activity is caused by the redispersion of Cu NPs into highly dispersed Cu²⁺ species, but we cannot tell whether the size or valence state dominates since the valence state of the Cu species is closely related to the particle size in our study as the referee said. We have conducted more experiments for evaluating catalytic performance (details can be seen in reply to the first referee) and modified the conclusion: “The redispersion of Cu NPs into highly dispersed Cu²⁺ species in O₂-H₂O contributes to the enhanced performance and recovered activity for RWGS and CO-PROX reactions.” on **Page 18** in the revised manuscript.

Reviewer #3: Fan et al. report on the effect of wet pretreatments of supported Cu nanoparticles on metal dispersion. While some interesting data are produced, I am not sure that the article meets the standards of Nature Communications.

Response: We really appreciate the reviewer for carefully reading the manuscript. After considering these suggestions, we have performed additional experiments and revised our manuscript carefully. We hope that the revision can satisfactorily address the reviewer's concerns. The following is the point-by-point response:

1. The introduction does not summarize the current knowledge on metal redispersion. It is more focused on a few examples dealing with Cu.

Response: Thank the referee very much for the valuable suggestion. We agree with the authors that the current knowledge and understanding of metal redispersion is highly needed in the introduction. Inspired by the referee, we summarize the current knowledge and commonly used methods for metal redispersion. Based on the discussions, we conclude that high temperature and specific atmosphere are usually required for the redispersion but it costs a lot. Our work provides an effective and energy-saving way to achieve redispersion of sintered metal catalysts. We have added the related discussions and updated the abstract and conclusion parts on **Pages 1 - 3, and 19** in the revised manuscript: “Supported metal nanocatalysts have been widely used in heterogeneous catalysis, while sintering of supported metal species is inevitable during high-temperature reactions leading to catalyst deactivation¹⁻⁵. Numerous redispersion strategies have been developed to reverse the sintering process and rejuvenate the active metal species, which are critical for chemical industries⁶⁻¹¹.....”

2. The main oxide supports, referred to as AlOOH-500 and -900, are claimed to be γ -Al₂O₃ (page 4), but there is no evidence for that phase.

Response: Thank the referee for the suggestion. XRD patterns (**Fig. R7**) confirm the γ -Al₂O₃ phase in AlOOH-500 and AlOOH-900, which have been added as **Fig. S1** in the revised SI.

Fig. R7 XRD patterns of AlOOH-T (T = 500/900). This is used as **Fig. S1** in the revised SI.

Sample	Shell	R (Å) ^a	CN ^b	ΔE_0 (eV) ^c	σ^2 (10^{-3}Å^2) ^d	R factor (%)
O ₂ -H ₂ O-4 h	Cu-O	1.94 (0.02)	3.0 (0.4)	2.1 (2.2)	3.7	1.1
O ₂ -4 h	Cu-O	1.96 (0.03)	2.0 (0.4)	9.8 (1.9)	5.4	1.8
	Cu-Cu	2.34 (0.05)	1.4 (0.5)	8.6 (2.3)	5.2	
Ar-H ₂ O-4 h	Cu-Cu	2.53 (0.08)	6.6 (0.6)	2.4 (1.4)	8.3	1.0
2Cu/AlOOH-900	Cu-Cu	2.51 (0.05)	7.0 (0.8)	0.5 (0.8)	8.2	1.7

3. Electron microscopy images in Figs. 1 and 4 show scale bars with no indication of the scale. In Fig. 1a, the supposed single Cu atoms are surrounded by circles that make any visualization impossible.

Response: Thank the referee for pointing out our errors. We have changed the marker from circles to arrows to make visualization clear. The modified **Fig. 1** is shown in the revised manuscript.

4. In Fig. 1f, EXAFS data are reported. However, neither XANES curves nor EXAFS fitting is provided. The components at R=4-5 Å are not ascribed for the Cu foil.

Response: Thank the referee for the constructive suggestion. XANES curves and fitting parameters of EXAFS are given in **Fig. S4** and **Table S1**. Our experimental results are consistent with the results in the literature, where the metallic Cu or Cu foil exhibits a bimodal peak at R = 4 - 5 Å. (*Nature*, 2023, 14, 262-269; *Nat Commun.*, 2020, 11, 3525)

Table S1. EXAFS distances and fitting parameters for the Cu-based catalysts. Fitting parameters: $S_0^2 = 0.72$ calculated using a Cu foil standard; k and R fit ranges are tabulated.

^aAtomic distance; ^bCoordination number; ^cDifference of potential between the sample and the standard; ^dDebye–Waller factor

Table R1. EXAFS distances and fitting parameters for the Cu-based catalysts. Fitting parameters: $S_0^2 = 0.88$ calculated using a Cu foil standard; k range: 2.3-11 Å⁻¹.

Sample	Shell	R (Å) ^a	CN ^b	ΔE_0 (eV) ^c	σ^2 (10 ⁻³ Å ²) ^d	R factor (%)
O ₂ -H ₂ O-4 h	Cu-O	1.94 (0.02)	3.0 (0.4)	2.1 (2.2)	3.7	1.1
O ₂ -4 h	Cu-O	1.96 (0.03)	2.0 (0.4)	9.8 (1.9)	5.4	1.8
	Cu-Cu	2.34 (0.05)	1.4 (0.5)	8.6 (2.3)	5.2	
Ar-H ₂ O-4 h	Cu-Cu	2.53 (0.08)	6.6 (0.6)	2.4 (1.4)	8.3	1.0
2Cu/AlOOH-900	Cu-Cu	2.51 (0.05)	7.0 (0.8)	0.5 (0.8)	8.2	1.7

5. Moreover, still for EXAFS, why only 2Cu/AlOOH-900 sample treated in O₂-H₂O is reported? One would at least expect the data for this sample before treatment.

Response: Thanks a lot for the important suggestions. Quasi *in-situ* EXAFS results of the fresh 2Cu/AlOOH-900 and samples treated with O₂/Ar-H₂O/O₂-H₂O for different times are shown in **Fig. R4**, which confirm the rapid redispersion into single atoms in O₂-H₂O, of which details can be seen in the reply to the first referee. The EXAFS results have been added as **Fig. 2b** and detailed discussions are given on **Pages 7, 8** in the revised manuscript: “Subsequently, quasi *in-situ* XAS experiments were conducted to identify the chemical state of Cu in O₂, Ar-H₂O and O₂-H₂O atmospheres.”

6. The XPS data in Fig. 2b are hardly visible, hidden by thick fitting curves. Moreover, why are the 2p_{1/2} components not shown?

Response: Thank the referee for the nice advice. We have decreased the thickness of the fitting curves to make them clear. Cu 2p_{1/2} components are added (**Fig. R2**). **Fig. 2c** is updated in the revised manuscript accordingly.

7. H-D exchange experiments: describe them and how they can be interpreted in terms of OH content, since nothing can be understood just from the claim “H-D exchange results show that Ar-H₂O treatment increases the OH content...”.

Response: Thank the referee for the nice suggestion. For H-D exchange experiment, the pretreated sample (Ar, 200 °C for 2 h) was exposed to D₂ with temperature increased from RT to 750 °C, and the change in the mass spectroscopy (MS) signals of HD (m/z = 3) was

recorded (*ACS Catal.*, 2023, 13, 2277–2285). According to the exchange reaction $D_2+OH\rightarrow OD+HD$, the HD signal can be used to detect the OH content on support surface (*ACS Catal.*, 2017, 7, 4083–4092).

We have added description of H-D exchange experiments including experimental details and mechanism for characterizing OH content on **Pages 8, 21** in the revised manuscript: “Our previous works have revealed that surface hydroxyl (OH) groups significantly affect the redispersion of metal NPs^{14,40}. According to the exchange reaction $D_2+OH\rightarrow OD+HD$, HD signal in the H-D exchange experiment can be used to detect OH content on support surface⁴¹.....” and “Typically, 0.1 g sample was loaded in a quartz tube and then pretreated under Ar atmosphere at 200 °C for 2 h. After the pretreatment, the H-D exchange experiment was started with a heating rate of 10°/min from RT to 750 °C by recording the mass spectroscopy HD signal ($m/z = 3$).”

8. Page 8, what is the “spontaneous monolayer dispersion phenomenon”? The way it is described, it looks more like coalescence than dispersion.

Response: Thank the referee for the question. The monolayer dispersion phenomenon was put forward by Prof. Youchang Xie in the last century. It holds that active components of many supported catalysts will disperse spontaneously onto the support surface as a monolayer or submonolayer (*Adv. Catal.*, 1990, 37, 1–43). According to the principle, no XRD diffraction patterns of supported species can be observed below the dispersion threshold (the amount of the active catalyst as a monolayer). Actually, monolayer-dispersed species are present as tiny atoms/clusters in most cases.

We have modified the discussion about the monolayer dispersion to make the concept clear on **Pages 11, 12** in the revised manuscript: **“Role of surface OH groups in the Cu dispersion.** For the spontaneous monolayer dispersion phenomenon proposed by Xie et al., active components will disperse onto support surface as a monolayer, which are present as tiny atoms/clusters in most cases and can’t be detected by XRD⁴⁵. The amount of active components as a monolayer is defined as the dispersion threshold, above which diffraction peaks of active components can be observed in XRD⁴⁶⁻⁴⁸.”

9. By the way, the use of the “dispersion” term throughout the paper is often improper.

“Dispersion” refers to the fraction of surface metal atoms in a nanoparticle. The process leading to single atoms or clusters from bigger entities (such as nanoparticles) is called “redispersion”.

Response: Thank the referee for the constructive suggestion. We agree with the referee that the term “redispersion” is more accurate to describe the phenomenon from bigger aggregates into smaller ones. We have replaced “dispersion” by “redispersion” in the revised manuscript.

10. Page 11, it is mentioned that XPS shows that hydroxylated Cu species “can” form on Si_3N_4 , but H-D exchange indicates that few OH groups are present on Si_3N_4 . However, it is concluded that surface hydroxylation is key to Cu redispersion. How does it account for XPS results?

Response: We really appreciate the reviewer’s constructive question. H-D exchange tests are conducted on pure support before and after O_2 - H_2O treatment, indicating that Si_3N_4 surface is difficult to be hydroxylated. XPS spectra are acquired over the supported Cu species before and after O_2 - H_2O treatment, suggesting Cu-OH species can be formed in O_2 - H_2O . Combining H-D exchange and XPS results, it can be inferred that the absence of redispersion on Si_3N_4 surface cannot be ascribed to Cu-OH species, and a high degree of surface hydroxylation (high OH content of support) is required for redispersion. Corresponding modifications have been made in the revised manuscript on page 14: “Quasi in-situ XPS results show that Cu-OH species can form on the 2Cu/ Si_3N_4 and 2Cu/h-BN samples after O_2 - H_2O treatment (Fig. S12a, b), and thus the different behavior may be caused by the supports rather than mobile Cu-OH species. H-D exchange experiments on pure supports indicate that O_2 - H_2O treatment significantly increases the OH content on OH-free h-BN surface while there are almost no OH groups exist on Si_3N_4 surface even after O_2 - H_2O treatment (Fig. 5e).”

11. Catalytic data (Fig. 6) look minimalist. For example, I would expect several successive heating/cooling cycles to assess the stability of the process. Would the two samples keep exhibiting different behaviors?

Response: Thank the referee so much for the professional suggestions. To demonstrate the

catalytic impact of this work, we have tested the activity and stability of RWGS and CO-PROX reactions, in which cyclic experiments were also conducted (Figs. R2, 8). The highly dispersed Cu species show high activity than Cu NPs in RWGS and CO-PROX reactions but would deactivate during reaction (stability test and heating/cooling cycles). The exposure of the deactivated catalysts to O₂-H₂O at RT can facily recover the activity, indicating that such regeneration method is practical for Cu-based catalysts (details can be also seen in the reply to the first referee). The new catalytic performance experiments have been added as Fig. 6 and Fig. S15 with corresponding discussions on Pages 16 - 18 in the revised manuscript and SI: “Effect of Cu redispersion on catalytic performance of RWGS and CO-PROX reactions. Due to the high CO selectivity and activity of copper, as well as its low cost compared to gold and platinum copper-based catalysts may be one of the most promising candidates for the RWGS reaction^{50,51}”

Fig. R8 CO₂ conversion of 2Cu/AlOOH-900 before and after exposure to O₂-H₂O for the first and second reaction cycles. Reaction condition: WHSV = 36000 mL/g_{cat}·h, 24%CO₂/72%H₂/4%N₂, P = 0.1 MPa. This is used as Fig. S15 in the revised SI.

12. The experimental data are only partial. For instance, plenty of parameters such as reactor diameter, flow rates, catalyst weight, etc., are omitted. Isn't the conversion formula (page 17) valid for CO oxidation?

Response: Thank the referee for the valuable suggestion. The conversion formula (new version on page 21 - 23) is also valid for CO oxidation. To make it clear, the detailed

experimental parameters and conversion formula for all reactions are updated on **Pages 21 - 23** in the revised manuscript: “Reverse water gas shift (RWGS) reactions were tested using a homemade fixed-bed micro-reactor with the weight hourly space velocity (WHSV) of 36000 mL/g_{cat}·h. The 50 mg pelleting catalysts (20 ~ 40 mesh) were loaded in a quartz tube with an inner diameter of 6 mm.”

13. Fig S8d: the flat background is surprising.

Response: We appreciate the reviewer's valuable comments. We repeated the H-D exchange tests on Si₃N₄ samples before and after O₂-H₂O treatment (**Fig. R9b**), and the results are consistent with those in the first manuscript (**Fig. R9a**), showing that Si₃N₄ surface is difficult to be hydroxylated. We also select another support (nano-BN) which is also not able to be hydroxylated and redispersion doesn't occur (**Fig. R10**). The results make us confirm that the hydroxylation of supports is crucial for redispersion. The results on nano-BN support are added as **Figs. S13h** and **14h** in the revised SI.

Fig. R9 Repeated H-D exchange results of Si₃N₄ before and after O₂-H₂O exposure. (a) in the first version and (b) in the revised manuscript. **Fig. R9b** is used as **Fig. 5e** in the revised manuscript.

Fig. R10 (a) XRD patterns of 2Cu/nano BN and (b) H-D exchange results of BN before and after O₂-H₂O exposure. This is used as **Figs. S14h** and **13h** in the revised SI.

14. Check “corporation” (page 3) and “conductive” (page 13) terms.

Response: We appreciate the question raised by the reviewer. We apologize for the spelling error in our original manuscript. The term “conductive” is right, of which the meaning is similar to “beneficial”. The “corporation” has been corrected to “incorporation” on **Page 3** in the revised manuscript.

Overall, due to the lack of important information as well as the poor analysis and presentation of the data, I do not recommend this paper for publication, at least in its present form.

Response: We appreciate the concern and constructive comments raised by the reviewer. We have carefully revised the manuscript and the following works have been done to address the referee’s concerns:

(1) The catalytic data are updated, adding RWGS and CO-PROX reactions to demonstrate that the highly dispersed Cu species generated in O₂-H₂O at RT is highly active in RWGS and CO-PROX reactions. Moreover, the stability test was conducted with several deactivation-regeneration cycles, which indicate that this work provides an effective and environmentally friendly method for the regeneration of sintered Cu-based catalysts.

(2) The EXAFS and XPS data are updated to provide more solid evidences supporting the

conclusion. In addition, we have re-analyzed the data and modified the corresponding discussions and forms of data presentation to enrich the understanding of underlying mechanisms.

REVIEWERS' COMMENTS

Reviewer #1 (Remarks to the Author):

I have read in detail the replies by the authors to my original questions and comments. Most of them are satisfactory to me and the manuscript has now been improved. Therefore, I support its publication in this journal.

Reviewer #2 (Remarks to the Author):

I have carefully read the authors' replies to the reviewers' comments and questions, and found all my concerns as well as those of the other reviewers have been properly handled. The readability and clarity of the manuscript also get largely improved after the revision. Therefore, I am happy to recommend the publishing of this paper in its current form. No further reviewing is needed.

Reviewer #3 (Remarks to the Author):

The manuscript, initially poor, has been quantitatively and qualitatively enriched following the referees' remarks. New experiments indicate the lack of stability of redispersed species. However, I guess that the paper is now acceptable for publication provided that the linguistic mistakes are corrected. Also check the EXAFS fitting parameters: "radial distance" and "energy variance" instead of "atomic distance" and "difference of potential"...

Reviewer #1: I have read in detail the replies by the authors to my original questions and comments. Most of them are satisfactory to me and the manuscript has now been improved. Therefore, I support its publication in this journal.

Reply: We really appreciate the referee for his/her review reports which definitely help us to significantly improve the quality of our manuscript.

Reviewer #2: I have carefully read the authors' replies to the reviewers' comments and questions, and found all my concerns as well as those of the other reviewers have been properly handled. The readability and clarity of the manuscript also get largely improved after the revision. Therefore, I am happy to recommend the publishing of this paper in its current form. No further reviewing is needed.

Reply: We sincerely appreciate the referee for the positive and constructive comments regarding our manuscript.

Reviewer #3: The manuscript, initially poor, has been quantitatively and qualitatively enriched following the referees' remarks. New experiments indicate the lack of stability of redispersed species. However, I guess that the paper is now acceptable for publication provided that the linguistic mistakes are corrected. Also check the EXAFS fitting parameters: "radial distance" and "energy variance" instead of "atomic distance" and "difference of potential"...

Reply: We thank the referee for his/her professional review work, constructive comments, and valuable suggestions on our manuscript. We have modified the description of "atomic distance" and "difference of potential" to "Radical distance" and "Energy correction".